# Methyl *t*-butyl ether-degrading bacteria for bioremediation and biocontrol purposes

**Giada d'Errico**[1], **Veronica Aloj**[1], **Valeria Ventorino**[1], **Assunta Bottiglieri**[1],
**Ernesto Comite**[1], **Alberto Ritieni**[2], **Roberta Marra**[1], **Sergio Bolletti Censi**[3], **Gavin R. Flematti**[4], **Olimpia Pepe**[1], **Francesco Vinale**[5,6]*

**1** University of Naples Federico II, Department of Agricultural Sciences, Portici (NA), Italy, **2** University of Naples Federico II, Department of Pharmacy, Naples, Italy, **3** Cosvitec scarl, Naples, Italy, **4** School of Molecular Sciences, The University of Western Australia, Crawley, WA, Australia, **5** University of Naples Federico II, Department of Veterinary Medicine and Animal Production, Naples, Italy, **6** National Research Council, Institute for Sustainable Plant Protection, Portici (NA), Italy

* frvinale@unina.it

**Data Availability Statement:** The 16S rRNA gene sequences obtained from bacterial strains were deposited in the GenBank nucleotide database under accession numbers from MN206777 to MN206791 (http://www.ncbi.nlm.nih.gov).

## Abstract

A total of fifteen potential methyl *t*-butyl ether (MtBE)-degrading bacterial strains were isolated from contaminated soil. They have been identified as belonging to the genera *Bacillus*, *Pseudomonas*, *Kocuria*, *Janibacter*, *Starkeya*, *Bosea*, *Mycolicibacterium*, and *Rhodovarius*. *Bacillus aryabhattai* R1B, *S. novella* R8b, and *M. mucogenicum* R8i were able to grow using MtBE as carbon source, exhibiting different growth behavior and contaminant degradation ability. Their biocontrol ability was tested against various fungal pathogens. Both *S. novella* R8b and *B. aryabhattai* were effective in reducing the development of necrotic areas on leaves within 48 hours from *Botritys cinerea* and *Alternaria alternata* inoculation. Whereas, *M. mucogenicum* effectively controlled *B. cinerea* after 72 hours. Similar results were achieved using *Pythium ultimum*, in which the application of isolated bacteria increased seed germination. Only *M. mucogenicum* elicited tomato plants resistance against *B. cinerea*. This is the first report describing the occurrence of bioremediation and biocontrol activities in *M. mucogenicum*, *B. aryabhattai* and *S. novella* species. The production of maculosin and its antibiotic activity against *Rhizoctonia solani* has been reported for first time from *S. novella*. Our results highlight the importance of multidisciplinary approaches to achieve a consistent selection of bacterial strains useful for plant protection and bioremediation purposes.

## Introduction

Methyl tertiary-butyl ether (MtBE) is widely used as an oxygenated additive to increase the octane number and the combustion efficiency of gasoline for reducing emissions of carbon monoxide and volatile organic carbon [1,2]. This chemical compound is commonly utilized for about 80% of motor vehicle fuels in the USA (10–15%) and in Europe (1–10%). Due to its high solubility (49 g L$^{-1}$), low organic carbon partition coefficient Koc (11 mg L$^{-1}$) and poor natural degradation, MtBE is highly persistent in the environment [3]. Therefore, MtBE is

**Funding:** This research was supported by MIURPON grant Marea 03PE_00106, POR FESR CAMPANIA 2014/2020- O.S. 1.1 grant Bioagro 559, MISE CRESO Protection n. F/050421/01-03/X32, PSR Veneto 16.1.1 Divine n. 3589659, and MIURPON grant Linfa 03PE_00026_1. The funders had no role in study design, data collection and analysis, decision to publish, or preparation of the manuscript. Cosvitec scarl provided support in the form of research materials and salary to SBC, but did not have any additional role in the study design, data collection and analysis, decision to publish, or preparation of the manuscript. The specific roles of these authors are articulated in the 'author contributions' section.

**Competing interests:** SBC is a salaried employee at Cosvitec scarl. There are no patents, products in development or marketed products to declare. This does not alter our adherence to PLOS ONE policies on sharing data and materials.

considered a serious environmental problem for the contamination of soil, surface water and groundwater [4,5]. Also, MtBE is a known animal carcinogen and a potential human carcinogen and genotoxin [4,6]. The development of technologies to treat MtBE-contaminated soil is of great importance worldwide. Different remediation technologies, such as soil flushing, soil washing, air stripping, adsorption, oxidation, phytoremediation, biodegradation processes and much more have been proposed [4,7]. Among these strategies, biodegradation processes are recognized as innovative, cost-effective and environmentally friendly options for the detoxification of MtBE-contaminated soil [4,7,8]. Some microorganisms can partially or completely degrade MtBE under aerobic or anaerobic conditions [9,10,11,12]. Hydrocarbon substrates in general are known to be selectively degraded by specific micro-organisms, consequently the use of microbial consortia can provide a broader spectrum of enzymes through co-metabolism [13]. A single microbial population can convert a compound into a metabolite that can be then degraded further by another population resulting in improved biodegradation [11]. Interestingly, several microbial agents tailored for bioremediation of contaminated soil are widely known for their ability to induce systemic resistance and in addition, promote plant growth. These include different genera and species of bacteria (*Bacillus*, *Streptomyces*, *Pseudomonas*, *Burkholderia* and *Agrobacterium*) or fungi (e.g. *Trichoderma*, *Talaromyces*) [14] as well as their metabolites [15,16]. The use of beneficial bacteria may be a mean of overcoming the problem of pollutants accumulated by soil and, at the same time, remediating polluted environments. The present work brings new insights on the biodegradability of MtBE by selected bacteria useful both for plant protection against various pests and diseases and for soil bioremediation, through investigations on: (1) MtBE degradation ability of bacterial strains isolated from hydrocarbon-contaminated soil; (2) *in vitro* and *in vivo* bio-control activity of selected bacterial strains; and (3) their induced systemic resistance (ISR).

## Materials and methods

### Culture media

The culture growth media used throughout this work were Potato Dextrose Agar (PDA), Potato Dextrose Broth (PDB), Luria Bertani Broth (LB), Luria Bertani Agar (LBA), Plate Count Agar, OXOID (PCA), Plate Count Broth, OXOID (PCB), and Malt Extract + Mycological Peptone (MEP). The SM (pH 6.6) contained (g·l$^{-1}$): $KH_2PO_4$, 0.68; $K_2HPO_4$, 0.87; KCL, 0.2; $NH_4NO_3$, 1; $FeSO_4$, 0.002; $ZnSO_4$, 0.002; $MnSO_4$, 0.002; $CaCl_2$, 0.2; $MgSO_4$, 0.2. Instead, Frankia-Tween (FTW) medium was comprised (g·l$^{-1}$) of the following: $K_2HPO_4$, 0.225; $KH_2PO_4$, 0.225; $(NH_4)2SO_4$, 0.225; $MgSO_4×7H_2O$, 0.05; $CaCO_3$, 0.005; $FeCl_2×4H_2O$, 0.005. The liquid mineral medium was sterilized by autoclaving at 121°C for 20 min. MtBE (Sigma-Aldrich, St. Louis, MO, USA; 99.9% purity) was added to media following sterilization and cooling at room temperature.

### Fungal microorganisms

Fungi used for assays were obtained from the fungal culture collection of the Department of Agricultural Sciences, University of Naples Federico II (Portici, Naples, Italy), and designated as: *Rhizoctonia solani*, *Pythium ultimum*, *Alternaria alternata* and *Botrytis cinerea*. Fungal inocula were produced by individually culturing of isolates for 2 weeks on PDA (for *A. alternata*), on MEP (for *B. cinerea*) or on PDB (for *R. solani* and *P. ultimum*) with shaking (150 rpm) in darkness at 25°C. Plates were flooded with sterile distilled water and gently rubbed with a sterile, bent glass rod to release conidia. The spore suspensions were decanted through pads of sterile glass wool to remove mycelial debris. The spore concentration was determined

using a Bright-line® hemocytometer (Sigma) and fungal constituents were cryopreserved with glycerol 20% (v/v) at -20°C until use.

## Isolation and identification of bacteria from MtBE-contaminated soil

**Microbial isolation.** MtBE-contaminated soil, collected from an industrial area in Italy, was used to isolate new potential hydrocarbon-degrading microorganisms. Briefly, 20 g of soil samples were suspendend in 180 mL of quarter strength Ringer's solution (Oxoid, Milan, Italy) containing tetrasodium pyrophosphate (16% w/v) according to Ventorino *et al*. [17]. After shaking, suitable tenfold dilutions were performed and used to inoculate (100 μL) PCA plates by using the Surface Spread Plate Count Method. Plates were incubated at 28°C for 48–72 h. After incubation, single colonies were randomly isolated on the basis of their colony morphology (i.e., shape, edge, color, elevation, and dimension) by repetitive streaking on the same isolation medium, and stored at 4°C as slant cultures until their characterization.

**Identification of bacterial isolates.** Bacterial isolates were identified using a polyphasic approach on the basis of their phenotypic, biochemical, and molecular characterization. Isolates were preliminarily characterized on the basis of their microscopic morphology (phase-contrast microscopy, shape, dimension, and presence of spores) and biochemical features (Gram stains and catalase activity) as previously reported [18].

Molecular identification was performed by 16S rRNA gene sequencing. Bacterial DNA was extracted using InstaGene™ Matrix (Bio-Rad Laboratories, Hercules, CA) according to the supplier's recommendations. Approximately 50 ng of DNA was used as template for PCR assay. The amplification of 16S rRNA gene was performed using synthetic oligonucleotide primers described by Weisburg *et al*. [19], fD1 (5′–AGAGTTTGATCCTGGCTCAG–3′) and rD1 (5′–AAGGAGGTGATCCAGCC–3′), *Escherichia coli* positions 8–17 and 1540–1524, respectively. The PCR mixture was prepared as reported by Alfonzo *et al*. [20]. The PCR conditions were performed as described by Viscardi *et al*. [21]. The PCR products were visualized by agarose (1.5% wt/vol) gel electrophoresis (100 V for 1 h) and then purified using the QIAquick gel extraction kit (Qiagen S.p.A., Milan, Italy). The DNA sequences were determined and analyzed as previously reported [22], and they were compared to the reference RNA sequences database of GenBank nucleotide data library using the Blast software at the National Centre of Biotechnology Information website (http://www.ncbi.nlm.nih.gov/Blast.cgi) [23].

## Selection of bacteria for soil bioremediation

**Screening in liquid medium.** Bacterial strains were pre-inoculated dissolving a single colony in 10 mL of PCB and incubated overnight at 30°C. After incubation, a volume of each culture, corresponding to 0.1 O.D._{600nm}, was used to inoculate 10 mL of FTW medium supplemented with 50 ppm of MtBE (Sigma-Aldrich) as the sole carbon source. The cultures were incubated at 25°C for 7 days and the growth of bacterial strains was determined by measuring the absorbance at 600 nm using a spectrophotometer (Eppendorf, Milan, Italy). All tests were performed in triplicate.

The bacterial strains able to grow in the selective FTW medium with MtBE were selected for further investigations. Pre-inoculum of selected bacterial strains was performed as above reported and used to inoculate 50 mL of FTW medium supplemented with different MtBE concentrations (50, 100 and 500 ppm). Cultures were incubated for 7 days at 25°C and 150 rpm, in a rotary shaker incubator. Samples were withdrawn periodically at 0, 3, 5, and 7 days and cell growth was determined by a spread plate count method using PCA medium.

**MtBE biodegradation assays.** The degradation of MtBE was monitored by Gas Chromatography-Flame Ionization Detection (GC-FID) using a Shimadzu GC-17 (Shimadzu, Kyoto,

Japan) instrument with autoinjector. Separation was achieved using a ZB-WAX column (60 m x 0.53 mm id, x 1.00 um phase thickness), split injection 1:25, injection time 2 min. The time programme was: 35˚C for 6 min, to 120˚C with 4˚C/min and to 240˚C with 40˚C/min, held at 240˚C for 5 min, injector and detector temperatures were 200 and 240˚C, respectively, helium (0.6mL/min) was used as a carrier gas. For headspace analysis, samples (10 ml) were stirred for 30 min at 70˚C in headspace vials containing 2.5 g NaCl. Gas from the headspace (1 µl) was injected into the GC-FID. For data acquisition and data processing GC Solutions software ver. 2.3 was used.

## Selection of bacteria for biocontrol

**Screening of bacteria for in vitro antagonistic activity towards fungal pathogens.** Selected bacteria, *B. aryabhattai* R1B, *M. mucogenicum* R8i and *S. novella* R8b, were tested for their ability to inhibit fungal soil pathogens (*R. solani* and *P. ultimum*) and foliar pathogens (*B. cinerea* and *A. alternata*) as described by Whipps [24]. Briefly, a single colony of each isolate was streaked on LBA plate and incubated for 3 days at 28˚C. Then, two single colonies of each isolate for strain were picked and placed in 500 µL of sterile water. Fifty microliter of each solution were streaked on PDA plate and incubated at 28˚C. After 24 h, a 5-mm agar disk containing actively growing margins of mycelial colonies was placed in the middle of the plate and incubated. Fungal growth inhibition was measured after 12, 24, 48 and 72 hours. Strains were tested in four replicates and the experiments were performed twice for each isolate. Untreated plates served as controls.

**Screening of bacteria for the in vivo biocontrol activity towards *B. cinerea* and *A. alternata*.** The ability of *S. novella* R8b, *M. mucogenicum* R8i and *B. aryabhattai* R1B to inhibit the growth of foliar pathogens, *B. cinerea* and *A. alternate*, was evaluated in *in vivo* assays. Tomato (*Lycopersicum esculentum* L. cv. Marmande) seeds were surface-sterilized (1% sodium hypochlorite for 2 min), rinsed several times, and sown in sterilized soil. After germination, the bacterial suspension was diluted with sterile distilled water to $1x10^7$ CFU/mL, and then immediately applied (3 mL) to tomato leaves using an aerosol spray bottle (Nalgene Inc., Rochester, NY). After foliar application, seedlings were treated with 10µL of fungal pathogen inoculum ($1x10^6$ conidia/mL) in germination buffer (20 mM potassium phosphate and 20 mM glucose). Plants were bagged and placed in a randomized block design and then incubated at 18˚C with >85% humidity for 7 days in a growth chamber (16 h/light photoperiod). Disease incidence was evaluated at 14 days after inoculation calculating the necrotic area per leaf ($mm^2$) for treatment. The experiment was repeated twice and each treatment was replicated four times. Untreated plants served as controls.

**Screening of bacteria for the in vivo biocontrol activity towards *P. ultimum*.** The ability of selected bacterial strains to inhibit the growth of soil pathogen *P. ultimum* was evaluated in *in vivo* assays. Tomato seeds cv. Marmande were surface-sterilized as described above. About one gram of seeds were exposed to 1 mL of each bacterial suspension ($1x10^7$ CFU/mL). After coating, seeds were sown in soil uniformly amended with the biomass of *P. ultimum* at a dose of 3 g/L per soil. The disease incidence was evaluated at 7 and 14 days after inoculation counting the number of germinated seeds and measuring root growth. Pots were placed in a randomized block design and then incubated at 22˚C under photoperiodic lighting (16 hours of light: 8 hours of dark) program (5,000 lux). The experiment included 8 treatments: untreated pots (water control), pots without bacterial and/or fungal inoculation (controls), and *P. ultimum*-infected pots treated with *B. aryabhattai* R1B, *M. mucogenicum* R8i and *S. novella* R8b. The experiment was repeated twice and each treatment was replicated four times.

**Induced systemic resistance (ISR) assays.** The capacity of *B. aryabhattai* R1B, *M. mucogenicum* R8i and *S. novella* R8b to induce systemic resistance in tomato plants cv. Marmande

against *B. cinerea* was tested. Tomato seeds were sterilized (as above described) and sown in sterilized soil. Three ml of each bacterial suspension were sprayed onto true-leaf stage of the first stand growth of test plants at approximately $1x10^7$ CFU/mL. Then, tomato leaves of the second stand growth were treated with 10μL of fungal pathogen inoculum ($1x10^6$ conidia/mL) in germination buffer. Plants were bagged and pots placed in a randomized block design at 18˚C under photoperiodic lighting (16 hours of light: 8 hours of dark) program (5,000 lux). Disease incidence was evaluated at 48, 72 and 96 hours after *B. cinerea* inoculation measuring the necrotic area per leaf ($mm^2$) for treatment. The experiment was repeated twice and each treatment was replicated four times. Untreated plants served as controls.

**Isolation and characterization of secondary metabolites produced by selected bacterial strains.** Selected bacterial strains were pre-grown in LB medium with shaking (150 rpm) at 25˚C for 2 weeks. Cultures were centrifuged (15000 rpm) for 15 min to remove bacterial cells. Then, liquid cultures of each strain were filtered through No. 4 filter paper (Whatman, Brentford, U.K.) and exhaustively extracted with ethyl acetate (EtOAc, Sigma-Aldrich, St. Louis, MO). The separated organic fractions were treated with anhydrous $NaSO_4$ (Sigma-Aldrich) to remove water moisture and evaporated *in vacuo* at 35˚C. The dried residue was subjected to analytical reverse-phase TLC (glass pre-coated Silica gel 60 RP-18 plates-Merck Kieselgel 60 TLC Silica gel 60 RP-18 F254s, 0.25 mm) using 8:2 v/v EtOAc: hexanes or 9:1 $CHCl_3$:MeOH as eluents. Compounds were detected on TLC plates using UV light (254 or 366 nm) and/or by spraying the plates with 10% (*w/v*) $CeSO_4$ in water or 5% (*v/v*) $H_2SO_4$ in EtOH followed by heating at 110˚ C for 10 min. The organic extracts obtained were submitted to silica gel column chromatography under atmospheric pressure (length 1.3 m and ø 4 cm). The sequence of elution step was performed using chloroform:methanol (8:2 v/v), chloroform:methanol (9:1 v/v) and methanol (MeOH 100%). Reactions were monitored by thin layer chromatography (TLC) using silica gel plates (Merck Silica Gel PF-254) and chloroform:methanol (9:1 v/v) as eluent. Homogeneous fractions were further purified by preparative TLC (Si gel; chloroform: methanol (9:1 v/v). All solvents and reagents used were supplied by Fluka (A.G. Bush, Svizzera).

**Antibiosis assays against four soil-borne pathogens.** The antibiotic properties of secondary metabolites extracted from selected bacterial strains were evaluated against soil-borne pathogens *P. ultimum*, *A. alternata*, *B. cinerea* and *R. solani*. Pathogen plugs from growing edges of colonies were placed at the center of Petri plates containing one-fifths of PDA. Ten microliters of the purified metabolite at concentrations ranging from 1 to 100 $\mu$g plug$^{-1}$ were applied on the top of each plug. The control was obtained by applying 10 μL of solvent alone (EtOAc). The solvent was evaporated under a laminar flow cabinet and plates were incubated at 25˚C for 3 days according to Almassi *et al* [25]. The pathogen growth was daily measured as colony diameter. Each treatment consisted of three replicates and the experiment was repeated twice.

## Statistical analysis

Statistical analysis was performed using SPSS 15.0 software (SPSS for Windows). As the results from the repeated experiments were similar, data were pooled for the analysis of variance (ANOVA). Means were compared using Student Newman Keuls multiple comparison test at $P < 0.05$.

## Accession numbers

The 16S rRNA gene sequences obtained from bacterial strains were deposited in the GenBank nucleotide database under accession numbers from MN206777 to MN206791 (http://www.ncbi.nlm.nih.gov).

# Results

## Identification and selection of bacteria isolated from MtBE-contaminated soil

A total of 15 potential MtBE-degrading bacterial strains were isolated from MtBE-contaminated soil after incubation on PCA medium. The polyphasic approach of identification resulted in bacterial isolates with different shapes, dimensions and, in some cases, spore presence and great biodiversity, as eight genera and twelve different species were found (Table 1). *Bacillus* spp. was the most representative genus with the species *B. aryabhattai*, *B. stratosphericus*, *B. thuringensis*, *B. mobilis*, and *B. marisflavi*. The other bacterial genera were represented by one species identified as *Pseudomonas xanthomarina*, *Kocuria rosea*, *Janibacter melonis*, *Starkeya novella*, *Bosea eneae*, *Mycolicibacterium mucogenicum*, and *Rhodovarius lipocyclicus* (Table 1).

Preliminary screening of all bacterial strains in the selective liquid medium showed that only the three strains *B. aryabhattai* R1B, *S. novella* R8b, and *M. mucogenicum* R8i, were able to grow in the FTW medium supplemented with MtBE (50 ppm) as sole carbon source (data not shown).

## Bacterial soil bioremediation

**Bacterial growth and MtBE biodegradation on liquid medium.** The three selected bacterial strains, *B. aryabhattai* R1B, *S. novella* R8b, and *M. mucogenicum* R8i, were tested for

**Table 1. Phenotypic characterization and molecular identification of 15 bacterial strains isolated from MtBE-contaminated soil.**

| Bacterial strains | Colony morphology | Cell morphology | Gram reaction | Catalase activity | Identification (% identity) | Accession Number |
|---|---|---|---|---|---|---|
| R1B | Opalescent, irregular | Rod-shaped, endospore-forming | + | + | *Bacillus aryabhattai* (99%) | MN206777 |
| R1C1 | White, irregular | Rod-shaped, endospore-forming | + | + | *Bacillus stratosphericus* (99%) | MN206778 |
| R2b | White, translucent, round | Rod-shaped | - | + | *Pseudomonas xanthomarina* (99%) | MN206779 |
| R3a | Pink-orange, circular, slightly convex, smooth | Cocci | + | + | *Kocuria rosea* (99%) | MN206780 |
| R4a | White, irregular | Rod-shaped, endospore-forming | + | + | *Bacillus thuringensis* (99%) | MN206781 |
| R7b1 | White, irregular | Rod-shaped, endospore-forming | + | + | *Bacillus mobilis* (99%) | MN206782 |
| R7C | White, irregular | Rod-shaped, endospore-forming | + | + | *Bacillus mobilis* (99%) | MN206783 |
| R7e2C2 | Yellow, irregular | Rod-shaped, endospore-forming | + | + | *Bacillus marisflavi* (98%) | MN206784 |
| R7e2C11 | White, round, convex | Cocci | + | + | *Janibacter melonis* (98%) | MN206785 |
| R7e2d | Yellow-orange, irregular | Rod-shaped, endospore-forming | + | + | *Bacillus marisflavi* (98%) | MN206786 |
| R8b | White-yellow, opalescent, round | Short rods | - | + | *Starkeya novella* (99%) | MN206787 |
| R8e | White-cream, smooth, round | Short rods | - | + | *Bosea eneae* (99%) | MN206788 |
| R8i | White, translucent, smooth, round | Rod-shaped | + | + | *Mycolicibacterium mucogenicum* (99%) | MN206789 |
| R8fa | White, round | Short rods | - | + | *Rhodovarius lipocyclicus* (99%) | MN206790 |
| O3a | White, irregular | Rod-shaped, endospore-forming | + | + | *Bacillus thuringensis* (99%) | MN206791 |

their ability to grow in the FTW medium supplemented with different concentrations of MtBE. Although all the strains were able to grow in the minimal selective medium containing different MtBE concentrations reaching values of $10^7$–$10^8$ in 3–5 days, a different behavior in respect to chemical compound concentration was observed. In particular, the strain *B. aryabhattai* R1B reached a concentration of approximately $10^8$ CFU/mL at 50 and 100 ppm of MtBE and showed the lowest growth on plates loaded with 500 ppm of MtBE (about $10^7$ CFU/mL) exhibiting a moderate ability to degrade MtBE at 50 ppm (46%), 100 ppm (37%) and 500 ppm (15%) (Table 2). Similarly the highest cell growth of the strains *M. mucogenicum* R8i and *S. novella* R8b was observed in the cultural medium at lower concentration of MtBE (50 ppm), reaching values of about $10^8$ CFU/mL (data not shown), exhibiting a significant degradation potential in the amount of 74 and 87%, respectively (Table 2). A lower growth, up to approximately $10^7$ CFU/mL (data not shown) on plates with 100 and 500 ppm of MtBE, was observed for both strains with a degradation percentage ranging from 14 to 0% (Table 2).

## Bacterial biocontrol activity

**Screening of bacteria for in vitro antagonistic activity towards fungal pathogens.** Antagonism was evaluated in terms of reduction of fungal radial growth. *Starkeya novella* R8b and *B. aryabhattai* R1B were effective in reducing *B. cinerea*, *P. ultimum* and *A. alternata* activities. Overall, strain R8b produced greater inhibition than strain R1B against *B. cinerea*. The inhibition caused by strain R8b was 50%, 39% and 24% for *B. cinerea*, *P. ultimum* and *A. alternata*, respectively (Fig 1). Whereas, strain R1B inhibited fungal radial growths of *P. ultimum* (25%) and *A. alternata* (32%). No sign of growth inhibition of *B. cinerea* occurred using the bacterial strain R1B (Fig 1). *Mycolicibacterium mucogenicum* R8i was not effective in reducing fungal activities. None of the selected bacterial strains inhibited *R. solani* radial growth (data not shown).

**Screening of bacteria for the in vivo biocontrol activity towards *B. cinerea* and *A. alternata*.** *Starkeya novella* R8b and *B. aryabhattai* R1B were effective in reducing the disease incidence caused by *B. cinerea* (Fig 2) and *A. alternata* (Fig 3). *Starkeya novella* R8b and *B. aryabhattai* R1B statistically reduced foliar damages caused by both pathogens after 48 hrs in comparison to the control. The inhibition caused by strain R1B and R8b was up to 50% and 85%, respectively, for *B. cinerea* in comparison to the control (Fig 2). Whereas, Fig 4 shows the fungal inhibition caused by strain R1B (ranging from 53% to 74%) and R8b (ranging from 60% to 76%) against *A. alternata*. No sign of growth inhibition of both tested pathogens occurred using the bacterial strain R8i (Figs 2 and 3). *Mycolicibacterium mucogenicum* R8i was not effective in reducing fungal activities. However, all selected bacterial strains were ineffective against *R. solani* (data not shown).

**Screening of bacteria for the in vivo biocontrol activity towards *P. ultimum*.** Results indicated that in absence of *P. ultimum*, bacterial-treated seedlings showed a more developed

**Table 2. Quantification of MtBE degradation (%) produced by *Bacillus aryabhattai* R1B, *Mycolicibacterium mucogenicum* R8i and *Starkeya novella* R8b in the presence of MTBE at 50, 100 and 500 ppm after 14 days of incubation.** Data were obtained using Gas Chromatographic-Flame Ionization Detector (GC-FID) method.

| Bacterial Strains | MtBE concentration (ppm) | | |
|---|---|---|---|
| | 50 | 100 | 500 |
| *B. aryabhattai* R1B | 46% | 37% | 15% |
| *M. mucogenicum* R8i | 74% | 4% | 11% |
| *S. novella* R8b | 87% | 14% | 0% |

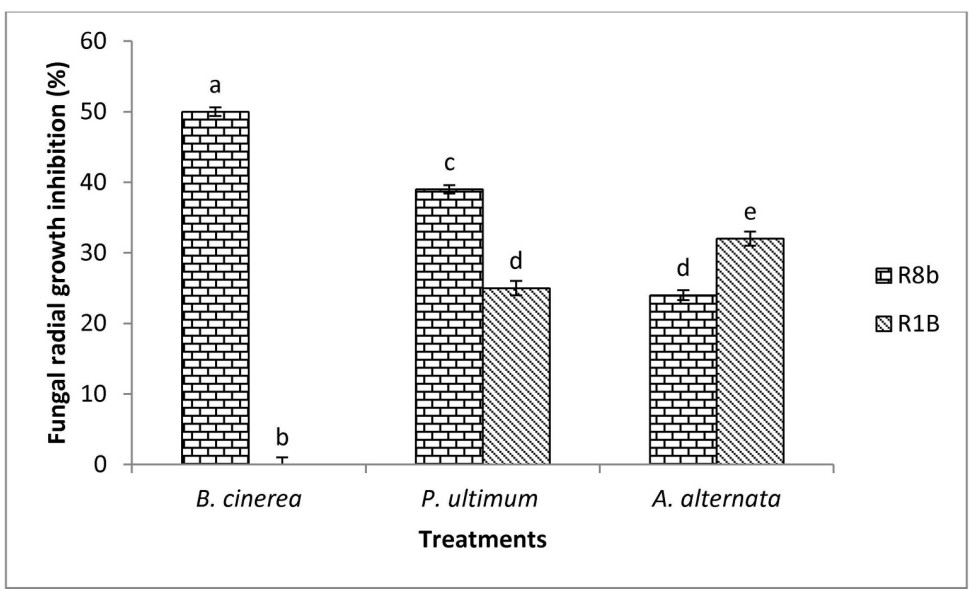

**Fig 1. Percent fungal radial growth inhibition of *Botrytis cinerea*, *P. ultimum* and *A. alternata* in response to *Starkeya novella* R8b (R8b) and *Bacillus aryabhattai* R1B (R1B) on Petri plates of potato dextrose agar (PDA) at 28˚C.** Inhibition area was measured after 96 hours of incubation. Values represent means of quadruplicate samples ± standard deviation. Means were compared using Student Newman Keuls multiple comparison test at P < 0.05. Different letters indicate significant difference between treatments (P < 0.05).

root system than the untreated control (Fig 4). In *P. ultimum*-inoculated soil, bacterial strains were not effective on tomato root growth (Fig 4). Whereas, seed germination was positively improved by treatments using all selected bacteria (Table 3).

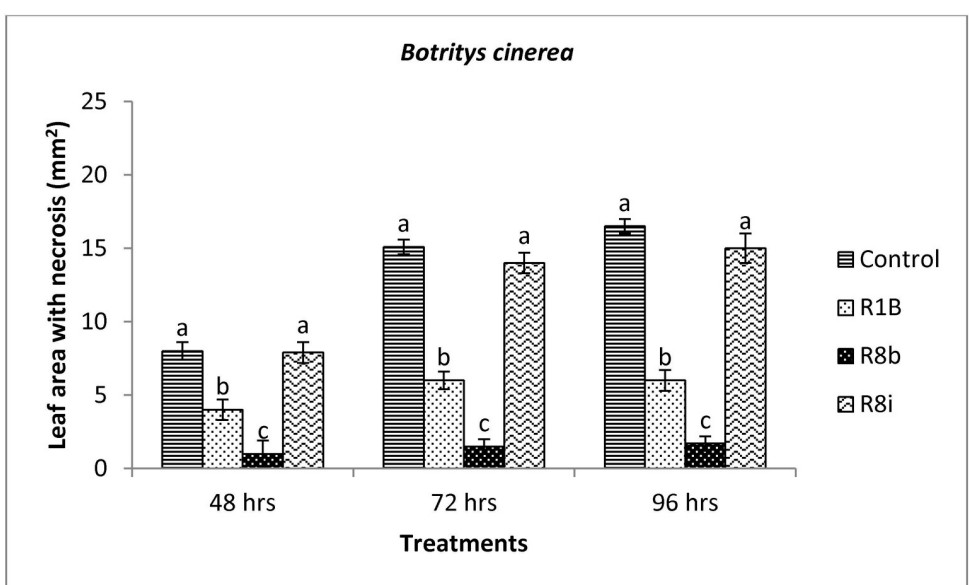

**Fig 2. Mean inhibition of necrotic leaf area (mm²) of *Botritys cinerea* in response to *Bacillus aryabhattai* R1B (R1B), *Mycolicibacterium mucogenicum* R8i (R8i) and *Starkeya novella* R8b (R8b) treatments.** Diameters of the necrotic area on leaves were measured after 14 days. Values represent means of quadruplicate samples ± standard deviation. Means were compared using Student Newman Keuls multiple comparison test at *P* < 0.05. Different letters indicate significant difference among treatments at 48, 72 or 96 hours (*P* < 0.05).

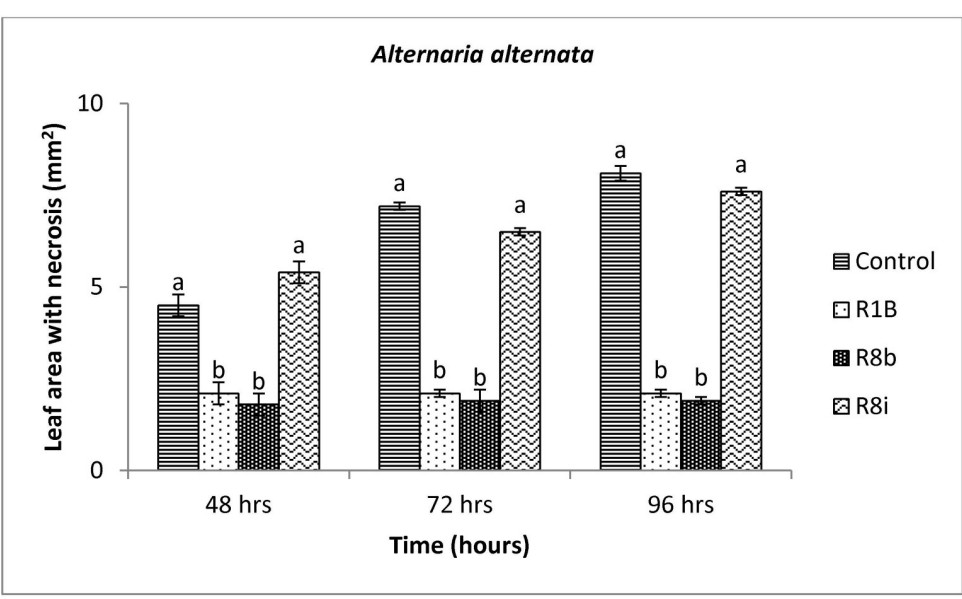

**Fig 3. Mean inhibition of necrotic leaf area (mm²) of *Alternaria alternata* in response to *Bacillus aryabhattai* R1B (R1B), *Mycolicibacterium mucogenicum* R8i (R8i) and *Starkeya novella* R8b (R8b) treatments.** Diameters of the necrotic area on leaves were measured after 14 days. Values represent means of quadruplicate samples ± standard deviation. Means were compared using Student Newman Keuls multiple comparison test at $P < 0.05$. Different letters indicate significant difference among treatments at 48, 72 or 96 hours ($P < 0.05$).

**Induced systemic resistance (ISR) assays.** When tomato plants were inoculated with *M. mucogenicum* R8i, the appearance of necrotic cell death after leaf infection with *B. cinerea* was reduced by 50% 48 hours after pathogen challenge (Fig 5). After 72 and 96 hours strain R8i

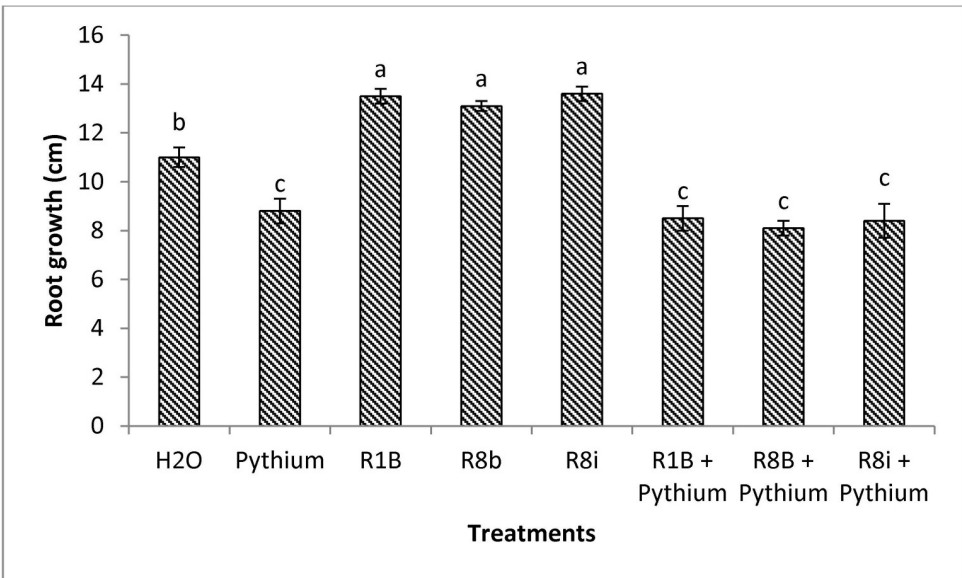

**Fig 4. Tomato root growth infected by *Pythium ultimum* and treated with selected bacterial strains.** H₂O (water), *Pythium ultimum*, *Bacillus aryabhattai* (R1B), *Starkeya novella* (R8b) and *Mycolicibacterium mucogenicum* (R8i) served as controls. Values represent means of triplicate samples, and error bars represent standard deviation of the mean. Means were compared using Student Newman Keuls multiple comparison test at $P < 0.05$. Different letters indicate significant difference between treatments at 14 days ($P < 0.05$).

elicited systemic protection by more than 70%. *B. aryabhattai* R1B and *S. novella* R8b did not induce a resistance in tomato plants (not shown).

**Isolation and characterization of secondary metabolites produced by selected bacterial strains.** Secondary metabolites obtained from culture filtrates of *B. aryabhattai* R1B and *M. mucogenicum* R8i were isolated and characterized as above reported. Oil residues of strains R1B and R8i (234.2 mg and 253 mg, respectively) were mainly composed of fatty acids and lipocarbohydrates as determined by NMR analysis. The separation of *S. novella* R8b extract (265.6 mg) yielded 14 different and homogeneous fractions. The main secondary metabolite of *S. novella* R8b was purified by the preparative TLC of fractions (2–3), (3–5)III and (6–14)d. The metabolite isolated showed chromatographic and spectroscopic properties (NMR and MS) similar to those reported in literature [26] for a compound known as maculosin (Fig 6).

**Antibiosis assays against four soil-borne pathogens.** *In vitro* antibiotic activity of the secondary metabolite maculosin against *R. solani*, *B. cinerea*, *P. ultimum* and *A. alternata* was tested. Results showed that only *R. solani* was slightly inhibited by maculosin at 100 μg (data not shown). Growth of *R. solani* was significantly reduced (up to 30%) by the highest concentration of maculosin after 48 hours of exposure.

## Discussion

Bioremediation is an innovative technology that employs the metabolic potential of the microbial soil component in the remediation of contaminated environments. Specialized bacteria are able to withstand unfavorable conditions and degrade specific pollutants such as hydrocarbons, heavy metals and various pesticides [27]. The involvement of these microorganisms in the bioremediation process have found wide application in environmental and agricultural sectors [28,29,30,31,32]. Numerous treatments exploit the opportunity to use pollutant-degrading microorganisms previously isolated from soils contaminated by the same compound [33,34]. In fact, recently, it was demonstrated the capacity of natural ecosystems to develop a microbiota adapted to polluted soil due to anthropogenic activities as release of organic xenobiotic compounds [35]. In this way, it is possible to discover well-adapted microorganisms that are potentially able to metabolize organic pollutants converting them into less toxic and/or less mobile products. Bioremediation includes different processes that could be combined or improved through genetic manipulation or by altering the physico-chemical conditions of polluted sites [31,36,37,38,39,40]. These strategies are obviously subject to legal and

**Table 3. Tomato seed germination after 7 and 13 days from *Pythium* inoculum.** $H_2O$ (water), *Pythium*, *Bacillus aryabhattai* (R1B), *Starkeya novella* (R8b) and *Mycolicibacterium mucogenicum* (R8i) served as controls. Values represent means of triplicate samples ± standard deviation. Means were compared using Student Newman Keuls multiple comparison test at $P < 0.05$. Different letters indicate significant difference between treatments at 7 and 14 days ($P < 0.05$).

| Treatments | Seed germination (%) | |
|---|---|---|
| | 7 days | 14 days |
| water | 90.4 a | 99.5 a |
| *Pythium* | 60.0 c | 65.3 c |
| R1B | 60.2 c | 97.9 a |
| R8B | 91.3 a | 98.9 a |
| R8i | 82.1 b | 96.9 a |
| R1B + *Pythium* | 80.3 b | 80.6 b |
| R8B + *Pythium* | 72.3 c | 81.9 b |
| R8i + *Pythium* | 82.2 b | 91.3 b |

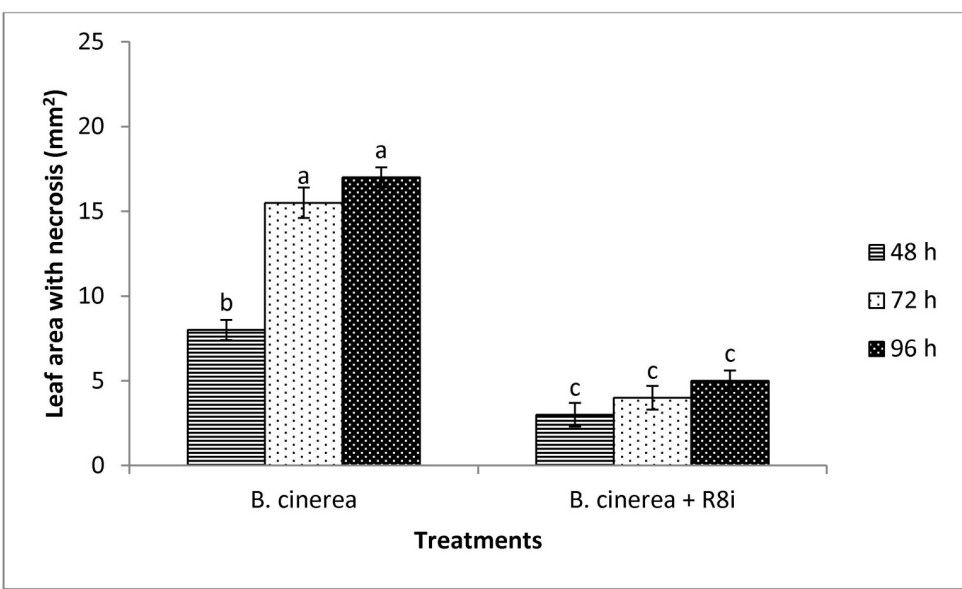

**Fig 5. Effect of treatments with *Mycolicibacterium mucogenicum* R8i (R8i) on tomato leaves cv. Marmande infected with *Botrytis cinerea*.** For induced systemic resistance (ISR) test, the true-leaf stage of the first stand growth of plants was sprayed with R8i ($1\times10^7$ CFU/ml) and the second stand growth was infected with *Botrytis cinerea* ($1\times10^6$ conidia/ml). Plants were bagged and incubated at 18˚C. Disease incidence was evaluated at 48, 72 and 96 hours, measuring the necrotic area per leaf (mm$^2$). The experiment was repeated twice and each treatment was replicated four times. Means were compared using Student Newman Keuls multiple comparison test at $P < 0.05$. Different letters indicate significant difference between treatments at 48, 72 or 96 hours ($P < 0.05$).

socio-political barriers. In addition to bioremediation, microbes have applications in other areas of biotechnology, including the biological control of plant diseases.

In recent decades, there is increasing interest in the biological control techniques of plant diseases because of the traditional chemical control is causing serious problems, not only at the environmental, but also sanitary and economic levels. The use of pesticides contributes to soil and water pollutions, produces resistant pests and interferes with beneficial microflora and/or microfauna. Thus, eco-friendly strategies for plant protection represent a valid alternative to the use of synthetic chemicals, more respectful of environment, animal and human health

**Fig 6. Structure of maculosin.**

[41,42,43,44]. Numerous species of bacteria are used in the biological control of plant diseases [45,46,47,48,49].

In view of these issues, bacteria play a key role in the bioremediation processes since they are able to degrade the organic matter producing a multiplicity of enzymes; moreover, they are characterized by a reproduction rate generally higher than other microorganisms [50]. In this work, 15 potential MtBE-degrading bacterial strains were isolated from MtBE-contaminated soil. Among these, the three strains *B. aryabhattai* R1B, *S. novella* R8b and *M. mucogenicum* R8i were able to grow using MtBE as carbon source, although they exhibited different growth behavior on high MtBE exposure as well as in degrading the chemical compound. Although the ability of some strains belonging to *Mycobacterium* genus to be involved into degradation of MtBE it was reported [51,52], this is the first known report describing the occurrence of this activity in *M. mucogenicum* species. Similarly, no previous works reported the ability of strains belonging to *B. aryabhattai* and *S. novella* species in degrading MtBE. However, *Bacillus* genus includes species that are able to degrade a wide variety of organic materials [53]. Recently, Wahla *et al.* [54] used a strain belonging to the *B. aryabhattai* species, isolated from contaminated soils, in consortium with other bacterial strains for biodegradation of the herbicide Metribuzin. It was reported that *B. aryabhattai* strains are also useful in arsenic bioremediation [55] as well as are able to promote rice seedlings growth and alleviate arsenic phytotoxicity [56]. Moreover, this species as well as *S. novella* are known to be able degrading organophosphate insecticides [57,58]. Dudášová *et al.* [59] reported a newly isolated bacterial strain *S. novella* with PCB-degrading ability in liquid medium as well as in PCB-contaminated sediment.

The biocontrol ability of selected bacteria was tested for the first time through *in vitro* and *in vivo* antagonism and ISR tests against various fungal pathogens. *In vitro* the most interesting results were obtained from *S. novella* R8b and *B. aryabhattai* R1B. These strains were effective in reducing the development of necrotic areas on leaves within 48 hours from the inoculation of *B. cinerea* and *A. alternata*. On the other hand, *M. mucogenicum* R8i effectively controlled *B. cinerea* after 72 hours from pathogen inoculation. Similar results were achieved using *P. ultimum*, in which the application of isolated bacteria increased seed germination. In our experiments, only *M. mucogenicum* R8i elicited tomato plants resistance against *B. cinerea*. The main mechanism of inhibition is due to the production of antibiotic compounds [60].

Although *M. mucogenicum* showed bioremediation and biocontrol proprieties, mycobacteria are dangerous human and animal pathogens, causing not only tuberculosis, leprosy and severe mycobacterioses [61]. With regard to mycobacterial diversity, investigations have revealed the presence of specific species for hydrocarbon-contaminated soils such as *M. monascense* and *M. chlorophenolicum* [62]. In particular, *M. mucogenicum* has been detected in the water and aerosol samples in a hospital therapy pool environment [63].

The secondary metabolite, obtained from *S. novella* R8b and identified as the diketopiperazine maculosin, is known as a host-specific fungal phytotoxin produced by *A. alternata* on *Centaurea maculosa* [26]. Thus, maculosin is considered as a chemical lead for developing an environmentally friendly antiknapweed herbicide [64]. However, we report for the first time that maculosin is also produced by *S. novella* and exhibit antibiotic activity against *R. solani*. These findings open interesting perspectives on the possibility of using bacterial microorganisms and/or their derivatives for the formulation of new commercial products for plant protection.

In conclusion, bioformulations are less dangerous than synthetic pesticides for human and animal health, and the environment. Our results highlight the importance of complementary screening steps through a multidisciplinary approach to obtain a more representative selection of bacterial strains for plant protection and bioremediation purposes.

## Author Contributions

**Conceptualization:** Olimpia Pepe, Francesco Vinale.

**Data curation:** Giada d'Errico, Veronica Aloj, Valeria Ventorino, Assunta Bottiglieri, Ernesto Comite, Roberta Marra, Olimpia Pepe, Francesco Vinale.

**Investigation:** Giada d'Errico, Veronica Aloj, Valeria Ventorino, Assunta Bottiglieri, Ernesto Comite, Alberto Ritieni, Roberta Marra, Gavin R. Flematti, Olimpia Pepe, Francesco Vinale.

**Writing – original draft:** Giada d'Errico, Veronica Aloj, Valeria Ventorino, Sergio Bolletti Censi, Olimpia Pepe, Francesco Vinale.

**Writing – review & editing:** Giada d'Errico, Veronica Aloj, Valeria Ventorino, Sergio Bolletti Censi, Olimpia Pepe, Francesco Vinale.

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
