## [Decision Letter · Decision Letter 0]

26 Nov 2019

PONE-D-19-29309

Methyl t-butyl ether-degrading bacteria for bioremediation and biocontrol purposes

PLOS ONE

Dear Dr Vinale,

Thank you for submitting your manuscript to PLOS ONE. After careful consideration, we feel that it has merit but does not fully meet PLOS ONE’s publication criteria as it currently stands. Therefore, we invite you to submit a revised version of the manuscript that addresses the points raised during the review process.

The paper has been revised by two experts who suggested to insert some minor revision in order to render the ms acceptable for publication in PlosOne.

We would appreciate receiving your revised manuscript by Jan 10 2020 11:59PM. To enhance the reproducibility of your results, we recommend that if applicable you deposit your laboratory protocols in protocols.io, where a protocol can be assigned its own identifier (DOI) such that it can be cited independently in the future. For instructions see: http://journals.plos.org/plosone/s/submission-guidelines#loc-laboratory-protocols

We look forward to receiving your revised manuscript.

Kind regards,

Sabrina Sarrocco

Academic Editor

PLOS ONE

Journal Requirements:

The authors received no specific funding for this work.

We note that one or more of the authors are employed by a commercial company: Cosvitec scarl.

Reviewers' comments:

Reviewer's Responses to Questions

**Comments to the Author**

1. Is the manuscript technically sound, and do the data support the conclusions?

Reviewer #1: Yes

Reviewer #2: Yes

2. Has the statistical analysis been performed appropriately and rigorously? 

Reviewer #1: Yes

Reviewer #2: Yes

3. Have the authors made all data underlying the findings in their manuscript fully available?

Reviewer #1: Yes

Reviewer #2: Yes

4. Is the manuscript presented in an intelligible fashion and written in standard English?

Reviewer #1: Yes

Reviewer #2: Yes

5. Review Comments to the Author

Reviewer #1: Article entitled “Methyl t-butyl ether-degrading bacteria for bioremediation and biocontrol purposes” by G. d´Errico and coworkers, described the isolation of a total of 15 strain from MtBE contaminated soils, from where they finally selected three strains able to grow in MtBE as sole carbon source. Authors study their ability to degrade MtBE, but also they analyze their in vitro and in vivo ability to antagonize three different fungal pathogens, as well as their effect in tomato seeds germination and tomato root growth in plants infected or not with the pathogen Pythium ultimum. Authors finally isolated the metabolite maculosin from strain R8b and determine their effect on inhibition of R. solani.

Article it is interesting since identified bacterial strains putatively interesting in bioremediation of MtBE contaminated soils, but also analyzed their ability to induce growth and defense response in plants, which both are together very interesting properties. Thus, this work includes new insights in these two, apparently unrelated fields, bioremediation and biocontrol.

The experiments are clearly presented and article well written.

I only have included some minor suggestions (below) that could contribute to the improvement of this manuscript.

Abstract

Lines 22-23. Please specify if B. aryabhattai refers to a Bacillus or a Bosea species.

Results

Section starting on Line 277.- It would be interesting to know why authors do not include strain M. mucogenicum R8i in this analysis. Please explain.

Legend to Fig. 2 must be wrong since in that Figure there are included data from strains R1B, R8b and R8i, while in the legend indicates that only data from B. aryabhattai R1b are included. Please correct the legend.

Section starting on line 337. Authors would have to explain more in detail why they only used strain M. mucogenicum R8i in this study.

Reviewer #2: The MS is overall well written, and authors presented using a multidisciplinary approach novelty results, reporting for the first time the occurrence of bioremediation and biocontrol activities of some species of Bacteria.

I suggest, if the authors had some good pictures of the experiment in vivo, to insert them in the MS or in the Supplementary Materials.

If know, what is the effect of these selected bacteria sprayed on the tomato leaves against some insect parasites, such as aphids, mites, and Tuta absoluta? Could be used also as insecticides?

If know, what is the effect of these selected bacteria against plant parasitic nematodes? Could be used also as nematocides?

If know, what is the impact of these selected bacteria on soil microorganisms such as collembola, mites and other micro arthropods?

I have some suggestions to improve the article:

Line 22 and lines 278, 295, 296: at the beginning of sentence report the entire name of the genus. Bacillus and not the abbreviation. Starkeya and not the abbreviation. Check it in all the MS.

Lines 58-60: I suggest to insert e.g. after the first bracket and to delete etc at the end.

Line 84: I suggest to change with culturing the isolates.

Line 93: Specify where the contaminated soil was collected.

Lines 149-157: Where is the control? It is not clear what bacterial strains were used. Please, clarify.

Lines 159-171: Where is the control? It is not clear what bacterial strains were used. Please, clarify.

Lines 186-196: Where is the control? It is not clear what bacterial strains were used. Please, clarify.

Line 434-436: …screening steps through a multidisciplinary approach to obtain…

6. PLOS authors have the option to publish the peer review history of their article (what does this mean?). If published, this will include your full peer review and any attached files.

Reviewer #1: No

Reviewer #2: No

---

## [Author Response · Author response to Decision Letter 0]

24 Jan 2020

Dear Editor,

MANUSCRIPT: PONE-D-19-29309

We are very grateful to the editor and reviewers for their suggestions, which have been very helpful in improving the manuscript. The revised manuscript with track changes and an unmarked version have been uploaded as separate files. We have made a considerable effort to consider all the interesting suggestions and corrections proposed by the reviewers. However, should it be necessary, further comments and suggestions are welcome. Below, are the point-to-point answers to the comments of the reviewers.

On behalf of all co-authors,

Yours sincerely,

Francesco Vinale

Reviewer comments:

Reviewer #1: 

1) Article it is interesting since identified bacterial strains putatively interesting in bioremediation of MtBE contaminated soils, but also analyzed their ability to induce growth and defense response in plants, which both are together very interesting properties. Thus, this work includes new insights in these two, apparently unrelated fields, bioremediation and biocontrol. The experiments are clearly presented and article well written.

Thank you for your positive comments.

2) Abstract (lines 22-23): Please specify if B. aryabhattai refers to a Bacillus or a Bosea species.

B. aryabhattai refers to a Bacillus species. This information is now indicated in the text.

3) Results: section starting on Line 277. It would be interesting to know why authors do not include strain M. mucogenicum R8i in this analysis. Please explain.

M. mucogenicum R8i was not effective in reducing fungal activities. This information is now indicated in the text.

4) Legend to Fig. 2 must be wrong since in that Figure there are included data from strains R1B, R8b and R8i, while in the legend indicates that only data from B. aryabhattai R1b are included. Please correct the legend.

Legend has been corrected.

5) Section starting on line 337. Authors would have to explain more in detail why they only used strain M. mucogenicum R8i in this study.

We have not used only M. mucogenicum R8i. Data are shown in figure only for this bacteria because B. aryabhattai R1B and S. novella R8b did not induce a resistance in tomato plants as reported at lines 346-347. 

 Reviewer #2:

The MS is overall well written, and authors presented using a multidisciplinary approach novelty results, reporting for the first time the occurrence of bioremediation and biocontrol activities of some species of Bacteria.

Thank you for your positive comments.

I suggest, if the authors had some good pictures of the experiment in vivo, to insert them in the MS or in the Supplementary Materials.

We have some pictures of the experiment in vivo but the quality is not high. If it is necessary we could insert them as supplementary materials. 

If know, what is the effect of these selected bacteria sprayed on the tomato leaves against some insect parasites, such as aphids, mites, and Tuta absoluta? Could be used also as insecticides?

If know, what is the effect of these selected bacteria against plant parasitic nematodes? Could be used also as nematocides?

If know, what is the impact of these selected bacteria on soil microorganisms such as collembola, mites and other micro arthropods?

Our work was aimed at the evaluation of these selected bacteria against fungal pathogens. Actually, we don’t have results against other organisms but future studies could investigate the activity of selected bacteria against other target organisms. 

Line 22 and lines 278, 295, 296: at the beginning of sentence report the entire name of the genus. Bacillus and not the abbreviation. Starkeya and not the abbreviation. Check it in all the MS.

All suggested changes are now included in the text.

Lines 58-60: I suggest to insert e.g. after the first bracket and to delete etc at the end

Suggested changes are now included in the text.

Line 84: I suggest to change with culturing the isolates.

In our opinion, the sentence is correct and do not needs this change.

Line 93: Specify where the contaminated soil was collected.

The contaminated soil was collected in proximity of a fuel distributor located in Campania Region (Italy).

Lines 149-157: Where is the control? It is not clear what bacterial strains were used. Please, clarify.

The text was modified according to your suggestions.

Lines 159-171: Where is the control? It is not clear what bacterial strains were used. Please, clarify.

The text was modified according to your suggestions.

Lines 186-196: Where is the control? It is not clear what bacterial strains were used. Please, clarify.

The text was modified according to your suggestions.

Line 434-436: …screening steps through a multidisciplinary approach to obtain…

This sentence was modified according to your suggestion.

---

## [Editor Report · Decision Letter 1]

28 Jan 2020

Methyl t-butyl ether-degrading bacteria for bioremediation and biocontrol purposes

PONE-D-19-29309R1

Dear Dr. Vinale,

We are pleased to inform you that your manuscript has been judged scientifically suitable for publication and will be formally accepted for publication once it complies with all outstanding technical requirements.

With kind regards,

Sabrina Sarrocco

Academic Editor

PLOS ONE
---

## [Editor Report · Acceptance letter]

14 Feb 2020

PONE-D-19-29309R1 

Methyl *t*-butyl ether-degrading bacteria for bioremediation and biocontrol purposes 

Dear Dr. Vinale:

I am pleased to inform you that your manuscript has been deemed suitable for publication in PLOS ONE. Congratulations! Your manuscript is now with our production department. 

With kind regards,

on behalf of

Dr Sabrina Sarrocco 

Academic Editor

PLOS ONE